# A Study of Aluminum Honeycomb Structures under Dynamic Loading, with Consideration Given to the Effects of Air Leakage

**DOI:** 10.3390/ma16062211

**Published:** 2023-03-09

**Authors:** Radosław Ciepielewski, Danuta Miedzińska

**Affiliations:** Faculty of Mechanical Engineering, Military University of Technology, Kaliskiego 2 Street, 00-908 Warsaw, Poland; radoslaw.ciepielewski@wat.edu.pl

**Keywords:** honeycomb, numerical modelling, dynamic compression, structure, air leakage

## Abstract

Aluminum honeycomb structures are used in the construction of protective materials due to the positive relationship between their mass and their energy-absorbing properties. Applying such materials in the construction of large machinery, such as military vehicles, requires the development of a new method of finite element modeling, one that considers conditions with high strain rates, because a material model is currently lacking in the available simulation software, including LS-DYNA. In the present study, we proposed and verified a method of numerically modeling honeycomb materials using a simplified Y element. Results with a good level of agreement between the full core model and the Y element were achieved. The obtained description of the material properties was used in the subsequent creation of a homogeneous model. In addition, we considered the influence of increases in pressure and the leakage of the air entrapped in the honeycomb cells. As a result, we were able to attain a high level of accuracy regarding the stress values across the entire range of progressive failure, from the loss of stability to full core densification, and across a wide range of strain rates.

## 1. Introduction

Functional, cell-oriented structures, including honeycombs, have a long history of successful application. For example, they were used in the initial period of space exploration, specifically in the Apollo 11 [1] project, to build elements of the crew and technical compartments. NASA and others [2,3,4] concluded that the honeycomb structure has the best energy-absorbing properties in relation to its specific mass. Due to this feature, honeycomb structures are used across a wide range of industries, including the automotive industry [5,6] and the military [7,8].

Sandwich structures with honeycomb cores are commonly used because of their high stiffness-to-weight ratio. Moreover, they have predictable progressive folding (stress–strain) characteristics in the case of static loading. This feature has made it possible to derive a mathematical model that enables the calculation of the so-called average value of crushing stress (plateau) to the core (occurring from the moment of stability loss to the beginning of the increase in force due to the structure’s densification) on the basis of the known, basic values characterizing honeycomb structures, such as the yield point of the core material, the wall thickness, and the cell size [9]. However, in [10,11], the authors show that the strength of the structure can increase by up to 50% with dynamic loading. In addition, there are a number of aspects that, if simplified, may make it impossible to effectively predict a honeycomb’s structural behavior, such as the air trapped inside the cells [12].

Modern tools, such as the finite element method (FEM) used in industry and science, allow for the effective prediction of the effects of dynamic loading, including fast-changing ones such as those that a shock wave from the detonation of an explosive may have on protective structures. In order to ensure a high-quality representation of the real conditions and the properties of the materials used, the data used in the preparation of the model should be as reliable as possible.

Scientific publications dealing with the numerical modeling of honeycomb structures can be divided into several groups, according to the subject covered by the authors. The most popular subjects include modeling the phenomenon of the loss of stability, modeling the local phenomena that occur during structural deformation, the determination of strength characteristics based on full geometric models and submodels, and observations of the phenomenon of energy absorption by structures modeled as homogeneous, using special constitutive models.

In [13], the authors focused on modeling the buckling process and progressive folding by comparing the effects obtained by using geometry to represent the samples, containing 16 single cells, used in their experimental tests, with results obtained from a four-cell compression analysis. The authors indicate that, by applying the appropriate boundary conditions—that is, by ensuring that the free edges of the walls maintained symmetry with respect to the plane perpendicular to them—the authors were able to obtain the same characteristics regardless of the number of individual cells that were modeled.

Similarly, in [14], the authors focused on determining the characteristics of dynamically loaded aluminum cores. The structures were modeled by retaining the full geometry corresponding to that which was used in the experimental research. Their test results show that successive walls inside the core deformed symmetrically in relation to their longer axes of symmetry.

In scientific nomenclature, the behavior of the core as a single-cell sector is referred to as the Y element. Among others, this modeling method was used in [15]. The authors presented the results of numerical analyses, the subject of which was the static compression of honeycomb cores as modeled in two different ways—by fully mapping the honeycomb geometry, and by solely modeling one single-unit sector. The main advantage of the method, in addition to the simple construction of the model and the significant reduction in the time needed to perform the calculations, is the ability to determine the full characteristics of the core, including the elastic range and the compaction range. The oscillations observed in the area of progressive folding represent a noticeable disadvantage.

A different approach to reducing the so-called numerical cost of the analysis of the strength of a cell core in an aluminum honeycomb is presented by the authors of [16], which is devoted to modeling the structure by using a structure consisting of joint-connected short beams. The authors used a Discrete Beam Method (DBM) and proved that the approach is correct and that it provides results that demonstrate a high level of agreement with those obtained experimentally.

Another approach often found in the literature [17,18]—which allows for a significant simplification of the metallic honeycomb geometric modeling process and a reduction of the time needed to perform the calculations—is the use of one of the constitutive models designed to reproduce the behavior of the loaded structure, which is modeled as homogeneous, usually using eight-node cubic elements. This method allows for a significant simplification of the core-modeling process, especially when it is applied to structures with large dimensions in relation to the characteristic dimensions of the structure. Modeling fine walls using elements with a small characteristic dimension, thus ensuring a reliable representation of the deformation of the core, reduces the time step in the case of dynamic analyses, leading to a significant increase in the time needed to perform the calculations.

In such cases, the MAT_HONEYCOMB constitutive material model, which is applied in FEM calculation systems (including LS-DYNA), can be used [19]. Its application allows for a full representation of the global stiffness of a structure, both axial and shear, in terms of elasticity, progressive folding, and complete compression, accompanied by its linear elastic characteristics. This model has been used to successfully predict the behavior of materials in which thin-walled, anisotropic-oriented structures were used, e.g., honeycombs, as well as in foamed metals and plastics. However, a significant simplification of the structural modeling process using MAT_HONEYCOMB requires the knowledge of a number of the parameters and characteristics of a particular core.

On the basis of the work presented in [20] produced by researchers at Toyota Motors^®^, the LS-DYNA system has been equipped with an additional constitutive model called MAT_MODIFIED_HONEYCOMB, which is designed to describe transversely isotropic bodies, the mechanical properties of which are symmetrical with respect to the axis normal to the isotropic plane. Honeycomb-type geometric structures have such a feature. However, industrially manufactured structures of this type are double-walled in one direction; therefore, treating them as transversely isotropic may be considered an oversimplification. Nevertheless, due to trends in the design of energy-consuming elements, including ones with regular spatial structures that have been created using additive manufacturing techniques, it is necessary to mention this material model as well. Because none of the material models described above have the ability to account for increases in stress caused by the pressure of compressed air trapped inside the cells, there is a need for a solution to that problem.

The aim of the present study was to develop and verify a method of modeling a homogenous honeycomb structure—as subjected to dynamic loads—using the characteristics obtained from a fast Y-element model. Such an approach has allowed us to obtain valuable results from FEM analysis, based on material models of the MAT_HONEYCOMB type, across a wide range of strain rates, thus minimizing the need for experimental research. The authors of [21] present experimental studies on the static and dynamic compression of honeycomb structures. The conclusion we drew from their work was that the strengthening of the structure is related to the volume of air trapped inside the closed cells.

## 2. Materials

Seven types of structures, all available from the catalogs of the manufacturers of these materials, were selected. The geometrical parameters of the individual structures are presented in Table 1. The name given to each structural type includes the following information: cell size–aluminum type–wall thickness. All dimensions used in the names are expressed in inches. An example of the studied material, along with the characteristic dimensions of its core, is shown in Figure 1.

## 3. Research Methodology

The research was conducted in the following stages:

First, a numerical analysis of the static compression of a simplified model of the honeycomb structure (the Y element) was completed with the introduction of imperfections related to the geometry of the real structure (based on computed tomography (CT) studies); validation of the model using experimental research was achieved.

Second, numerical analyses of the honeycomb structures were performed, using a homogeneous model and characteristics obtained from the calculations of the Y element, including the influence of the air trapped within the cells of the structure.

The scheme of the research stages is presented in Figure 2.

## 4. Models

In this section, the preparation of the model and its validation are presented. The conditions for the numerical analyses of static compression using the simplified Y element and those using the homogenous model are presented in Section 4.1 and Section 4.2, respectively.

### 4.1. Numerical Analyses of Static Compression Using the Simplified Y Element

The smallest, repetitive subarea that can be separated from a structure is a cell with the base of an equilateral triangle, the sides of which are perpendicular to the walls of three adjacent honeycomb cells. A separated single cell (the Y element) is shown in Figure 3.

As indicated in [13,14,15], structures consisting of repeating elements can be analyzed by solely focusing on a model that has been simplified down to the level of a segment, using appropriate boundary conditions. In the case of the analyzed structures, these conditions were limited to the treatment of the sides of the subarea as planes of symmetry.

The geometric model shown in Figure 4a, when prepared for discretization, consisted of three walls connected to each other in line with one of the longer edges. The walls were connected to each other at an opening angle of 120°.

Four-node, fully integral shell elements with five integration points across the element thickness were used to build the finite element mesh, as presented in Figure 4b. The characteristic dimension of the element was 0.23 mm, and there were 100 of them at the wall height. The selection of the element size was preceded by an analysis of the impacts that changing this parameter would have. At levels below the dimensions described above, no improvements were observed. The Belytschko–Tsay elements were used. To carry out the static compression analysis, all degrees of freedom were fixed in the top nodes and in the nodes next to the plane that forces motion (except that which was translational in the *Z*-axis). The boundary conditions, along with the symmetry conditions and the local coordinate systems, are shown in Figure 4c.

In order to accurately reproduce the actual structure and identify imperfections, the tested samples were subjected to computed tomography (CT) studies.

A SkyScan 1174 tomograph was used for this task. The obtained point clouds were processed to reduce noise, and then they were used to generate a polygonised structure; discrete, triangular surfaces were created between the closest points. In this way, spatial models of the cores were obtained, which was useful for making measurements. The axonometric views of the CT models for the 1/8-0.0015 and 3/16-0.0015 samples are shown in Figure 5.

Analyzing the actual structures of the cores via the creation of three-dimensional models using computed tomography allowed for the examination of the basic geometric parameters, e.g., the length of individual walls, the size of the cell, and the bending radii of the walls near the joints. Exemplary measurement results are shown in Figure 6.

The results of the geometric measurements, averaged for individual samples, are presented in Table 2.

On the basis of the data presented above, it can be seen that the double walls have a smaller width and the single walls have a higher width than those declared in the catalog cards. Moreover, the portion of the wall that is arched is a distinct part of it. The obtained measurement results allowed us to conclude that the bend radii of the walls depend mainly on their thickness, and that the dimensions are linked by a ratio of approximately 10:1.

In order to check the impact of the above-described geometric parameters, models of the structure of the sample designated as 1/8-0.0015 were built, one using a simple, idealized version, and the other constructed using the mapped shape of the walls according to the parameters described in Table 2. Both versions are shown in Figure 7, which demonstrates that, in addition to the visible rounding of the edges of the single wall (marked in red), the width of the double wall (marked in blue) has been reduced in favor of the single wall.

The introduced change in the geometry resulted in differences in the form of structural deformation that were apparent as early as the initial loss-of-stability phases. As can be seen in Figure 8, there are seven ridges on the flat core walls, and there are six smaller ridges on the rounded sides.

Minor differences in the geometry of the compared models were also reflected in the stress–strain characteristics obtained via the compression test. The value of the critical stress decreased by 11.6% (from 8.65 MPa to 7.64 MPa), and the value of the mean breaking stress decreased by 15.5% (from 5.98 MPa to 5.05 MPa). The value of the longitudinal modulus of elasticity before loss of stability also changed by 28.1% (from 3.11 GPa to 3.24 GPa). These are significant differences, thus the change in the model’s geometry should be taken into account in simulations. A comparison of the discussed characteristics, with reference to those obtained experimentally [21], is presented in Figure 9.

The constitutive model MAT_003_PLASTIC_KINEMATIC [19] was used to reflect the behavior of the material from which the tested Y element was made: aluminum alloy Al 5052-H39. It is a bilinear model, the stress–strain relationship of which is described by defining the tangent of the angle of the elastic part and the part of the kinematic strengthening. The parameters and mechanical properties of the material were taken from a report published by NASA [22] because the parameters of the thin, multi-rolled foil significantly differ from the parameters of the material from which it was made. In [22], the results of tests of the foil composed of the Al 5052-H39 alloy, as well as those of the foil fragments cut from the finished structure of the cellular core, are presented.

The basic parameters of aluminum Al 5052-H39, as used to make cores with a honeycomb topology in the untreated form [23], in the form of foil [22], and in the form of a fragment of the core [24], are presented in Table 3. After plastic processing, the material is characterized by significantly lower strength parameters. In the form of a film, it has a 19% lower Young’s modulus, a 30% lower yield point, and a 25% lower tensile strength. The sample cut from the core fragment has a Young’s modulus that is up to 46% lower, a 32% lower yield point, and a 35% lower tensile strength.

Due to the fact that the developed Y-element models were used to determine the mechanical properties of the statically compressed aluminum honeycombs without the need for experimental testing, the properties of the Al 5052-H39 foil were considered in the subsequent numerical calculations.

### 4.2. Numerical Analysis of the Honeycomb Structures Using the Homogenous Model

The aim of the tests described below was to assess the effectiveness of using the results of the simulation of the Y element in models that describe the global response of this type of material without a detailed analysis of the behavior of their internal structure.

Geometric models, as shown in Figure 5, were analyzed in order to identify fully closed cells and the possible damage resulting in the formation of open cells. Then, the volumes of the solids bounded by the closed walls of the cores were measured. Measurement results for the same types of samples were averaged. A visualization of an example of the discussed process is presented in Table 4. The samples with smaller core sizes had a noticeably higher proportion of closed-cell volume, considering that they had the same global dimensions.

The previous experimental research of honeycomb structures under various levels of strain rate loading carried out by the authors [21] was used to develop the assessments for this paper. By presenting the experimentally obtained, absolute increases in plateau stress [21] in the domain of the air volume locked in by the cell walls, an almost-linear characteristic was obtained, which can be observed in Figure 10. This observation may indicate that the main factor influencing the final shape of the stress–volumetric strain characteristic is the increase in the pressure of the air trapped inside the cells, especially for dynamic testing (strain rate of 3.8 × 10^2^ and 3.3 × 10^3^ 1/s).

Another argument for the correctness of this assertion is the conclusion made on the basis of the observation of images recorded with a high-speed camera in the dynamic test using a split Hopkinson pressure bar (3.3 × 10^3^ 1/s). As shown in Figure 11, the gas began to leak from the inside of the sample during the final compression phase (t = 333.30 µs). It became visible due to a dust cloud—a mixture of air escaping from the inside of the sample and fine particles of the damaged core and resin. This phenomenon took place in each case after reaching approximately 50% of the deformation; it continued after the maximum displacement of the initiating bar face, and this continued even after its plane lost contact with the sample plane. The described phenomenon may prove that, during the test, strong air compression takes place inside the core, which is released only after complete compression.

The above conclusion is important because none of the constitutive models available in commercially used computing environments take into account this kind of phenomenon’s influence on the change in the global mechanical properties of the structure.

Therefore, in order to describe the behavior of aluminum honeycomb structures in terms of strain rates higher than quasi-static, it is necessary to take an approach that will account for both the reaction of the deformed, thin-walled core and that of the air pressure inside the cells.

Assuming, as a simplification, that the air closed inside the cells is compressed without heat exchange, the increase in pressure ∆P can be described by a simple equation [25],
(1)$$\Delta P=P_{0}\left(\frac{V_{0}}{V}-1\right)$$
where *P*_0_ is the initial pressure (atmospheric).

Then, the indirectly measured stress value of the real structure would be the sum of the stress value in the structure (treating it as homogeneous according to the assumption made) and the pressure at a given time step. Due to the fact that the deformation of the core takes place only in the direction of displacement, the change in volume can be treated as a change in the height of the sample; as a result, the expected value of the measured stress is described by the formula [25]
(2)$$\sigma =\sigma_{r}+P_{0}\left(\frac{T_{0}}{T_{0}-u}-1\right)$$
where $\sigma_{r}$ is the stress in the core of the sample treated as a continuous medium, and $T_{0}$ is the initial core height.

The assumptions formulated above lead to the conclusion that the air trapped inside the cells causes an increase in the noted stress value, regardless of the strain rate. There are studies [24] which prove that the key aspect linking the structure’s response with the duration of its destruction are air leakages resulting from imperfections in the structure and the material discontinuities formed in the process of deformation. The assumption can be modified in accordance with research carried out by Xu et al. [24] and Hu et al. [25], in which it was shown that the cross-sectional area of a honeycomb block also changes during axial compression. The air leakage δ is determined by the relationship [21]
(3)$$\delta =1-\frac{PV}{P_{0}V_{0}}$$

Therefore, the pressure value can be determined by the formula
(4)$$P=P_{0}\frac{1-\delta }{1-\varepsilon_{v}}$$
where $\varepsilon_{v}$ is the volumetric strain.

It is assumed that *δ* is the core failure time of the *t*’ function and differentiating Equation (4) over time, assuming that one obtains
(5)$$\dot{P}=\frac{P_{0}}{1-\varepsilon_{v}}\left(\frac{1-\delta }{1-\varepsilon_{v}}\cdot \dot{\varepsilon_{v}}-\dot{\delta }\right)$$
and the leakage rate is
(6)$$\dot{\delta }=\frac{1}{P_{0}}\left[P\dot{\varepsilon_{v}}-\dot{P}\left(1-\dot{\varepsilon_{v}}\right)\right]$$

Based on the relations presented above, it can be noticed that the pressure value and the rate of the leakage are related to the strain rate $\dot{\varepsilon_{v}}$. Researchers developing analytical models for this type of issue [12], using the assumption that the strain rate is constant during the test, derived a relationship that links the pressure change inside the core with the strain and the leakage rate
(7)$$\Delta P=P_{0}\left(\frac{1}{1-\varepsilon_{v}}-1\right)\cdot \left(1-\frac{\dot{\delta }}{\dot{\varepsilon_{v}}}\right)$$

Equation (16) does not contain any unknown parameters other than the leakage rate. The authors of [25] state that this rate should be determined empirically by comparing the stress–volumetric strain results obtained during the compression of the samples with the cores tightly closed between covers, and those with openings releasing air during the test. The leakage intensity is very similar when the results of testing honeycomb structures with the same t/d ratio are compared, and its value depends on the strain rate. The obtained characteristics of the leakage intensity as a function of the strain rate for one of the cases analyzed in [25] (1/8-5052-0.001) is presented in Figure 12. In this case, the leakage rate increased almost in direct proportion to the value of the strain rate. This means that a tenfold increase in strain rate resulted in at least a tenfold increase in the leakage rate.

To perform simulation tests of the uniaxial compression of the homogenous honeycomb materials, a model with a cylinder geometry with a diameter of 25 mm and a height of 10 mm was used, reflecting the global geometry of the sample core. It consisted of 525 elements with a hexagonal topology.

The material model MAT_26_HONEYCOMB, available in the LS-DYNA system, was used to map the core, which was treated as a homogeneous material [19].

In MAT_26, in the uncompressed state (initial state, loss of stability, and progressive folding), the material retains its orthotropic properties, and the stress tensor components remain unconnected from each other so that the strain component in one local direction does not cause reaction forces in the others. Modules of the longitudinal and shear stiffness in particular directions depend on the given modules of the initial stiffness and the stiffness of a fully compressed (compacted) structure. These dependencies are as follow [19]:(8)$$E_{aa}=E_{aau}+\beta \left(E-E_{aau}\right)$$
(9)$$E_{bb}=E_{bbu}+\beta \left(E-E_{bbu}\right)$$
(10)$$E_{cc}=E_{ccu}+\beta \left(E-E_{ccu}\right)$$
(11)$$G_{ab}=G_{abu}+\beta \left(G-G_{abu}\right)$$
(12)$$G_{bc}=G_{bcu}+\beta \left(G-G_{bcu}\right)$$
(13)$$G_{ca}=G_{cau}+\beta \left(G-G_{cau}\right)$$
where
(14)$$\beta =\max\left[\min\left(\frac{1-V}{1-V_{f}},1\right),0\right]$$
and *E* is the modulus of elasticity of the core material; *G* is the shear modulus of the core material; $E_{aau}$, $E_{bbu}$, and $E_{ccu}$ are the modules of elasticity of the uncompressed cores; $G_{abu}$, $G_{bcu}$, and $G_{cau}$ are the shear modules of the uncompressed cores; V is the relative volume (the ratio of the current volume to the initial volume), and$\;V_{f}$ is the relative volume at which the core is considered fully compressed and it transforms into a linear elastic characteristic (the relative volume of total compaction).

In addition, the material model requires the definition of a set of curves that present the material characteristics obtained in the experimental tests: compression in each of the basic directions and shear in each of the base planes. There are two ways to define these characteristics. The first one is to determine the magnitude of stresses as a function of the relative volume (*V*). It is also possible to determine the magnitude of stresses as a function of volumetric strains, defined as
(15)$$\varepsilon_{V}=1-V$$

Finally, the components of the stress tensor are calculated according to the following relationship:(16)$$\sigma_{ij}^{n+1}=s_{ij}^{n+1}-p^{n+1}\delta_{ij}$$

After the process of updating the stress values is completed, they are converted into a global form.

For the modeling of individual structure types, the parameters presented in Table 4 were adopted. In each case, the *σ_ij_ – ε_ij_* characteristics were developed by selecting characteristic points from the curves obtained by the FEM simulation using the Y-element model.

The development of a model capable of taking into account the change in the nature of the response of the change of the initial conditions causing the strain-rate increase was based on one of the available methods describing the behavior of air-filled elements.

The model should allow us to describe the change in gas pressure, along with the change in the volume inside where it was located. It should also take into account the leakage in the calculation of the pressure changes and the application to the surface and the spatial boundary of the element. All of the possibilities mentioned above are offered by one of the simplest models: AIRBAG_SIMPLE_AIRBAG_MODEL [19]. The current value of the pressure acting on the boundaries of the vessel domain is calculated using the equation of state [19]:(17)$$P=\left(\gamma -1\right)\rho e$$
where *P* is the pressure, ρ is the density, and *e* is the internal energy of gas.

The γ coefficient is the adiabatic exponent: the ratio of specific heat at a constant pressure $c_{p}\;$to specific heat at a constant volume $c_{v}$ [25]
(18)$$\gamma =\frac{c_{p}}{c_{v}}$$

The rate of changes in the air mass *m* inside the volume, with time, is described by the relationship [26]
(19)$$\frac{dm}{dt}=\frac{dm_{i}}{dt}-\frac{dm_{o}}{dt}$$

The value of the air mass, which flows in via subsequent time steps *m_i_*, is defined by the appropriate characteristic. There are two options for determining the mass of the outflowing air *m_o_*: by defining the area of the holes through which the air leaks and their shape coefficient, and by defining the mass characteristics over time. There is also a possibility of making the size of the surface of the holes and the shape factor dependent on the value of the pressure inside.

In the discussed approach, the components of the energy balance [26]
(20)$$\dot{e}={\dot{e}}_{i}-{\dot{e}}_{o}-P\dot{V}$$
are as follows: ${\dot{e}}_{i}$ is energy change caused by the mass of inflowing gas, ${\dot{e}}_{o}$ is the change in energy caused by the mass of the outflowing gas, $P\dot{V}$ is work achieved by the pressure per volume change.

The described boundary condition was applied to the surfaces of the elements on the outer walls of the cylinder, as shown in Figure 13. This allowed for a direct transfer of the forces resulting from the increase in pressure to the sample boundaries. The parameters used to describe the air enclosed inside the sample are presented in Table 5. All parameters related to the influence of the air mass were omitted.

The initial boundary conditions are presented in Figure 14. An explicit scheme of the integration of the equations of motion was used for the calculations.

## 5. Results

The results are presented in two subsections in accordance with the descriptions of models presented in Section 3. The results of the numerical analyses for the quasi-static compression tests are described, first for the simplified Y element, and then for the homogenous model, in which the material characteristics gained via analyses of the Y element were implemented.

### 5.1. Results of the Numerical Analyses of the Static Compression of the Y-Element

The basic result of the performed analyses was the characteristic of the change in the stress value as a function of the volumetric strain and the list of resulting parameters. The characteristics obtained in the static and experimental FEM analyses [21], presented in Figure 15, reveal a consistent course of changes in the stress value, with an increase in the strain value. The similarity is especially visible in the elastic phase until it reaches critical stress, progressive folding, and compaction. The most pronounced difference in the stress values occurred just after the loss of stability. In each of the experimentally tested cases, a subsequent deep decrease occurred, up to about one-half of the subsequent progressive folding stress. The characteristics obtained by the FEM analysis did not include this feature. After the loss of stability, the force value decreased and stabilized until the compaction phase. Another effect that was recorded only during the simulation was the oscillation of the stress value in the progressive folding and compaction phases. The first of the causes mentioned above should be viewed as being involved with the local nature of such phenomena: as a loss of stability, the formation of a single fold, its closing, the mutual contact of walls, the subsequent loss of stability, etc. In the real structure, the local effects compensated for each other and there were no oscillations. Another noteworthy feature of the recorded oscillations of the stress value in the progressive folding phase was the increase in the amplitude with an increase in the wall thickness and a decrease in frequency with an increased cell size. In contrast, the oscillations in the densification phase were related to the occurrence of contact forces and friction due to the contact of the progressively larger surfaces of the walls with each other and their sliding over one another.

The obtained characteristics of the full range of deformations allowed for the identification and determination of the basic parameters of the individual types of honeycomb cores. Table 6 presents a summary and comparison between our results and those obtained through empirical research. Values included within parentheses denote the percentage of difference between those obtained through experimentation and those obtained through FEM analyses.

### 5.2. Results of the Numerical Analyses of the Honeycomb Structures Using the Homogenous Model

The stress–volumetric strain characteristics obtained through experimental tests [21] and those obtain via numerical analyses using the solid (homogenous) model are presented in Figure 16. Each of the graphs contain the results obtained during the tests carried out at three strain rates: 8.3 × 10^−2^, marked as QS; 3.8 × 10^2^, marked as DH; and 3.3 × 10^3^, marked as SHPB. The summaries are grouped according to the type of structure under study. In order to emphasize the characteristics obtained in the FEM calculations, the previously shown characteristics from empirical studies [21] are marked with dashed lines.

Characteristics obtained by means of the FEM analyses were consistent with their equivalents obtained experimentally, and the tendency to strengthen with an increase in the strain rate was maintained. Additionally, the nature of this tendency is similar; that is, in the plateau stress range, no hardening of a constant, proportional value occurred. It can therefore be assumed that the use of the material characteristics obtained via the static tests of the model—and the inclusion of the air enclosed inside the core under the test conditions in the model—brought about the intended results.

No vibrations occurred in any of the analyzed cases, and no other symptoms of wave phenomena were observed, which were recorded during the experiments using the split Hopkinson pressure bar. However, it should be mentioned that the method of determining the stress value was indirect and consisted of the measurement of the deformation of a long, slender bar that is also susceptible to oscillation in directions other than longitudinal. Therefore, the oscillations did not result from the response of the material being tested, but from the method of measurement and reading. The characteristics obtained in the FEM tests were smooth, devoid of sinusoidal components characterized by high amplitudes or low frequencies.

It should be pointed out that the characteristics obtained experimentally and numerically differ significantly in the initial range. While the elastic range had a similar course in all cases, as well as in terms of the nature of the stress build-up in the initial range, after reaching the value defined by the *σ_ij_ − ε_ij_* curve characteristic for the MAT_HONEYCOMB constitutive model used, the stress value stabilized; in the remaining further range of strain, it followed the values defined by it. Importantly, we observed no phenomenon where a clear plateau value was achieved that was much higher than that of the critical stress, followed (in the case of a real structure) by a loss of stability and the kind of breakdown of the structure’s stability, as manifested by a decrease in the stress value to a level below the plateau value that was achieved and maintained at a later stage in most of the compression tests. A more precise observation of this issue is shown in Figure 17, which presents the stress–volumetric strain characteristics of the 1/8-5052-0.0015 sample with the displayed range of volumetric strain limited to 20%.

## 6. Conclusions

The MAT_HONEYCOMB model does not include an option that allows for the consideration of the phenomenon by indicating a scalar stress value, up to which the material response would be dependent on the modulus of longitudinal elasticity, followed by assuming a value consistent with the indicated characteristic.

In order to obtain the full stress–strain characteristics of a metallic honeycomb, a numerical model can be used, the geometry of which will be limited to a single Y element. In order to obtain descriptions of the stress–volumetric strain characteristics and the scalar parameters, one must make sure that the following conditions are met. Regardless of the assumed strain rate, it is appropriate to apply the characteristics from static tests and achieve correct simulation results under a wide range of strain rates, assuming that the model takes into account the pressure change due to the potential compression of the air enclosed in the core volume.

In this study, the effectiveness of one of the simpler methods was tested: the definition of the boundary condition of uniform pressure distribution over a limited area. This enables the development of a model that faithfully reproduces the responses of a honeycomb structure at different strain rates.

## Figures and Tables

**Figure 1 materials-16-02211-f001:**
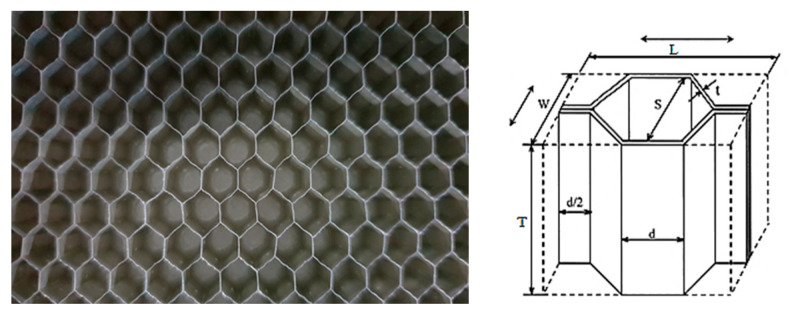
An example of the analyzed honeycomb structure (**left**), and the basic designations of the honeycomb’s structural elements (**right**) [9].

**Figure 2 materials-16-02211-f002:**
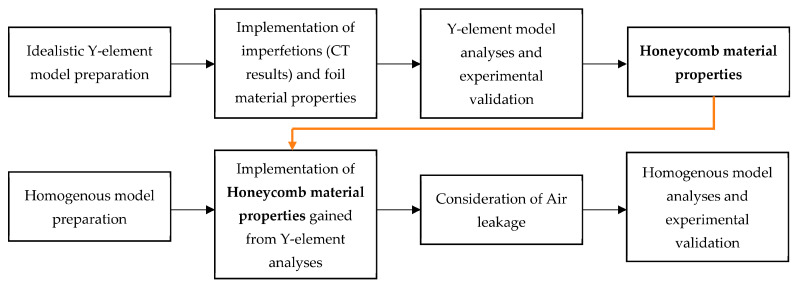
Research methodology scheme.

**Figure 3 materials-16-02211-f003:**
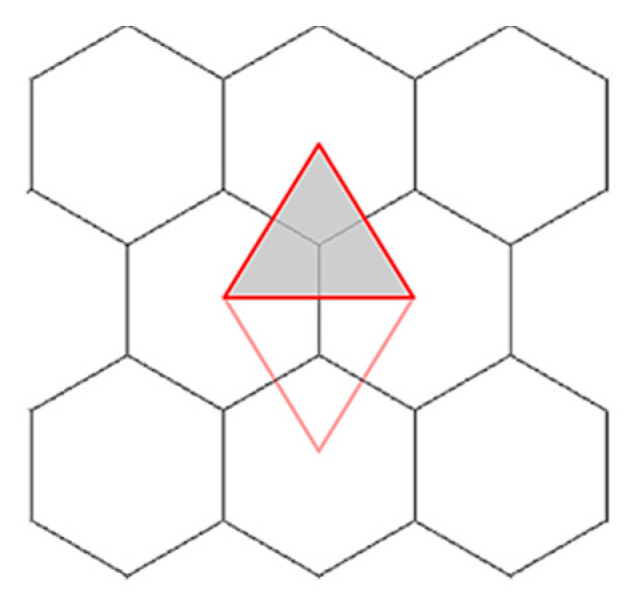
A single cell (Y element), as contained within a honeycomb structure.

**Figure 4 materials-16-02211-f004:**
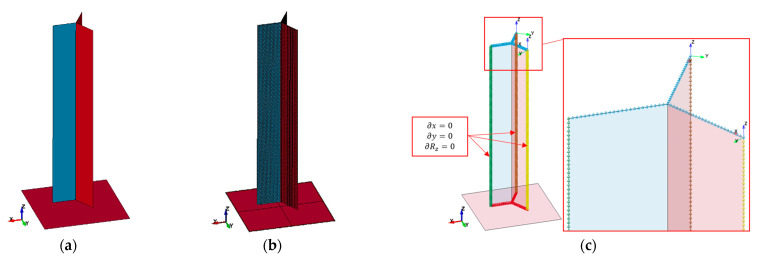
Y element: (**a**) geometric model, (**b**) finite element model, (**c**) boundary conditions.

**Figure 5 materials-16-02211-f005:**
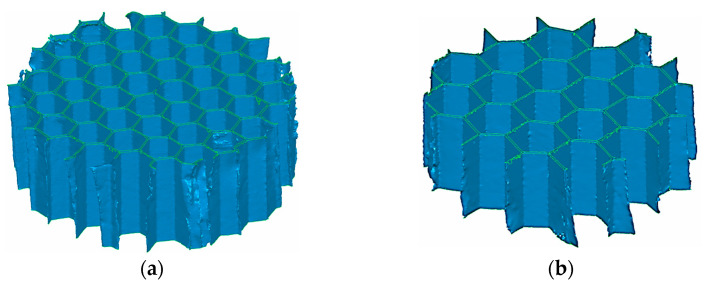
CT models of two of the studied materials: (**a**) 1/8-0.0015 and (**b**) 3/16-0.0015.

**Figure 6 materials-16-02211-f006:**
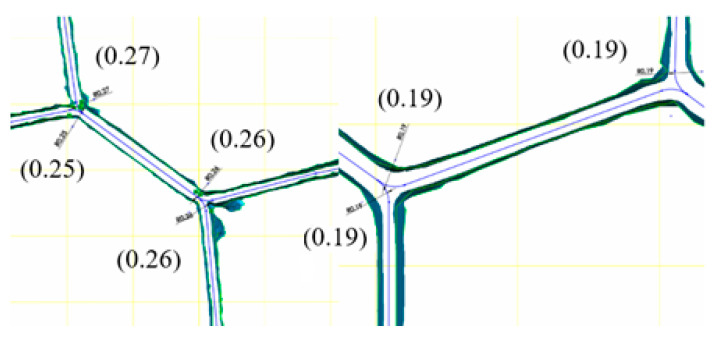
Examples of the measurement results of the geometrical parameters of the actual cores.

**Figure 7 materials-16-02211-f007:**
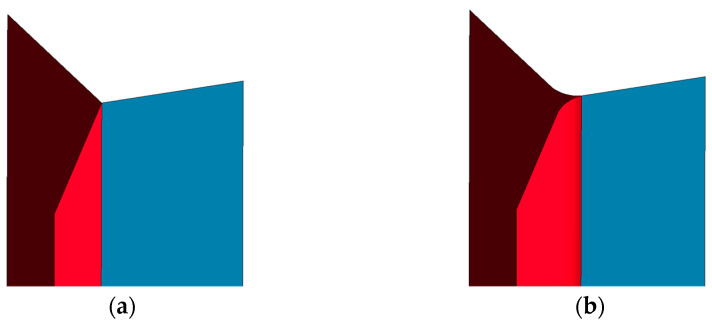
Geometric models of the core for the following versions: (**a**) simple/idealized; (**b**) showing the mapped shape of the rounded walls.

**Figure 8 materials-16-02211-f008:**
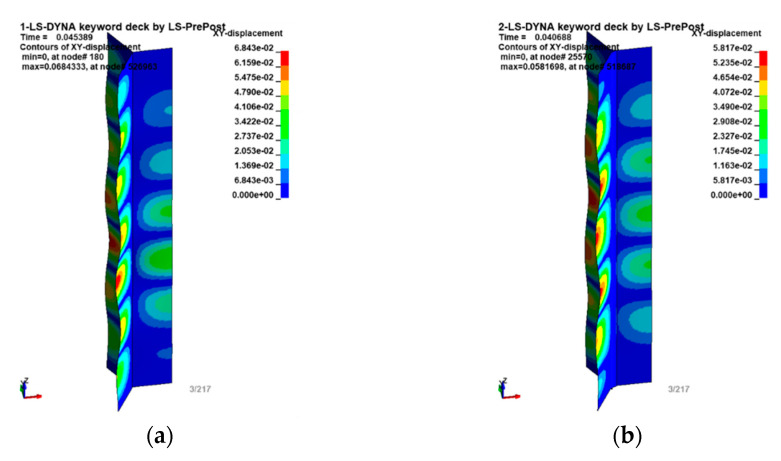
Maps of displacements [in mm] in the X–Y plane of the model in (**a**) the straight/idealized version; and (**b**) with the mapped shape of the rounded walls.

**Figure 9 materials-16-02211-f009:**
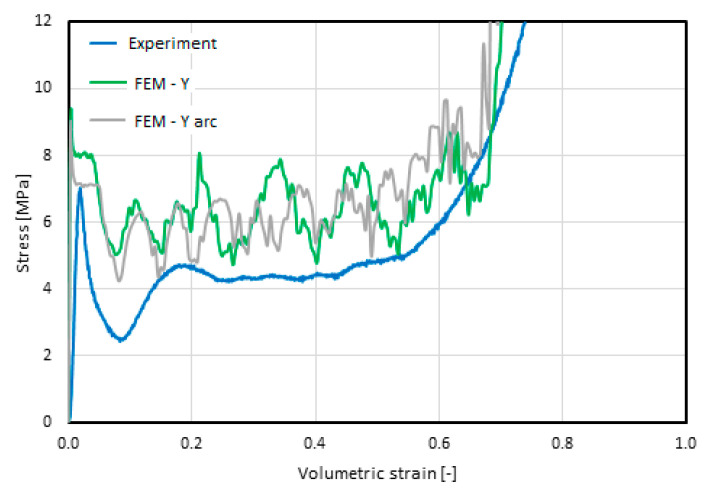
Comparison of the stress–volumetric strain characteristics of material 1/8-5052-0.0015 obtained for the following versions of the models: straight/idealized (FEM–Y, green line); with the mapped shape of the rounded walls (FEM–Y arc, gray line); and experimentally obtained characteristics, blue line [21].

**Figure 10 materials-16-02211-f010:**
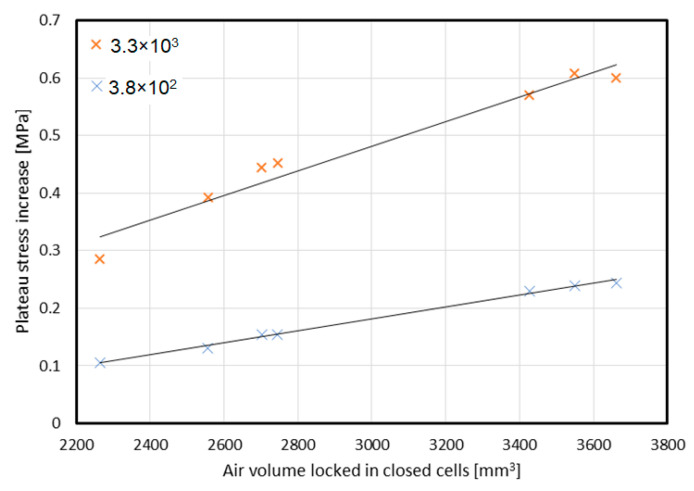
Increase in plateau stress depending on the volume of air trapped inside cells (based on results obtained in [21]).

**Figure 11 materials-16-02211-f011:**
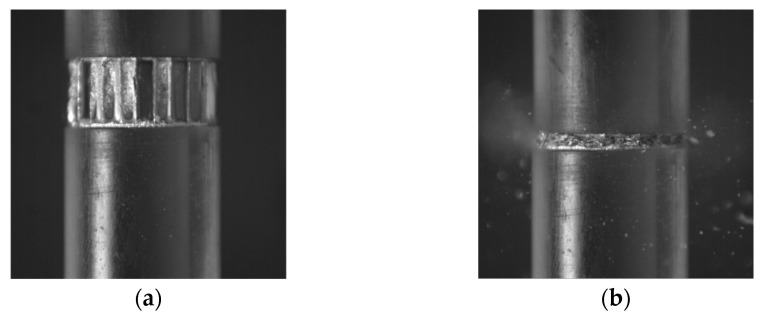
Deformation of sample 1/8-5052-0.0015 in the following time steps: (**a**) t = 0, and (**b**) t = 333.30 µs (based on results obtained in [21]).

**Figure 12 materials-16-02211-f012:**
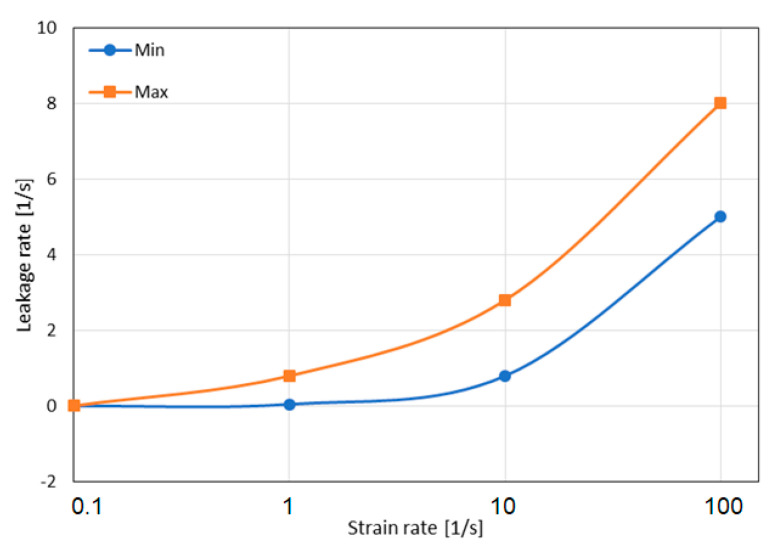
Leakage rate vs. strain rate for 1/8-5052-0.001 (based on [25]).

**Figure 13 materials-16-02211-f013:**
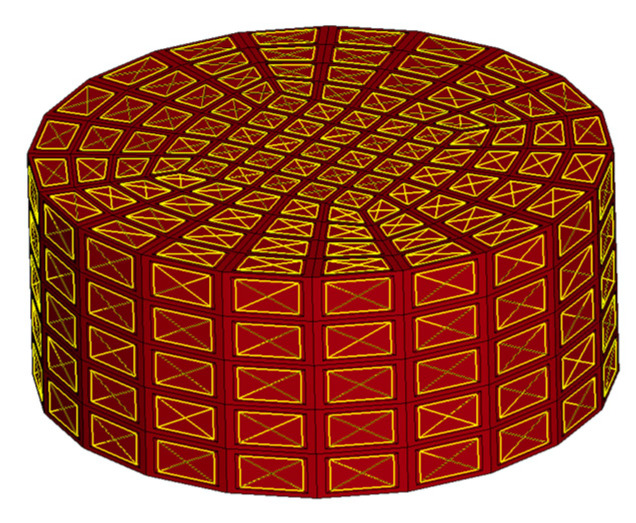
Numerical model of the homogeneous honeycomb material, with surfaces used as air-leakage boundaries.

**Figure 14 materials-16-02211-f014:**
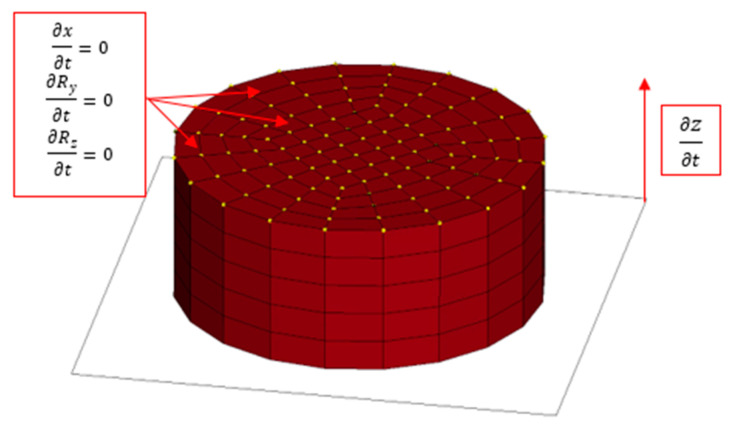
Scheme of the initial boundary conditions applied in the model.

**Figure 15 materials-16-02211-f015:**
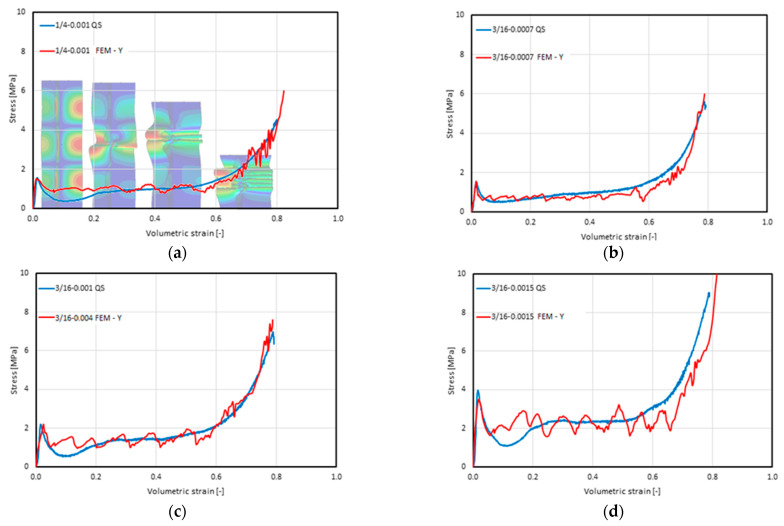
Summary of the stress–volumetric strain characteristics obtained using the FEM–Y model (FEM–Y) and experimentally in the quasi-static compression (QS) test [21] of the following material types: (**a**) 1/4-5052-0.001; (**b**) 16-5052-0.0007; (**c**) 3/16-5052-0.001; (**d**) 3/16-5052-0.0015; (**e**) 1/8-5052-0.0007; (**f**) 1/8-5052-0.001; and (**g**) 1/8-5052-0.0015.

**Figure 16 materials-16-02211-f016:**
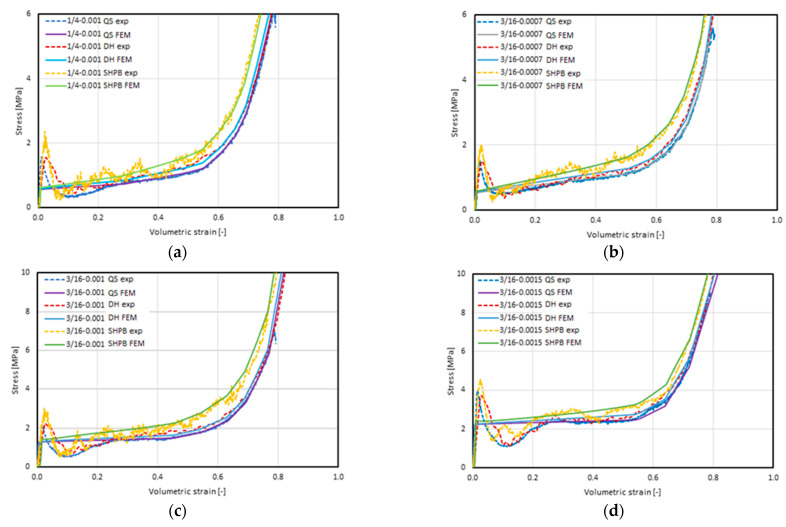
Summary of the stress–volumetric strain characteristics obtained using the FEM–homogeneous model (FEM) and experimentally (exp) [21] at a quasi-static load (QS) strain rate of 8.3 × 10^−2^ 1/s, at a dynamic load (DH) strain rate of 3.8 × 10^2^ 1/s, and using a Split Hopkinson Pressure Bar (SHPB) with a strain rate of 3.3 × 10^3^ 1/s) of the materials, including (**a**) 1/4-5052-0.001; (**b**) 3/16-5052-0.0007; (**c**) 3/16-5052-0.001; (**d**) 3/16-5052-0.0015; (**e**) 1/8-5052-0.0007; (**f**) 1/8-5052-0.001; and (**g**) 1/8-5052-0.0015.

**Figure 17 materials-16-02211-f017:**
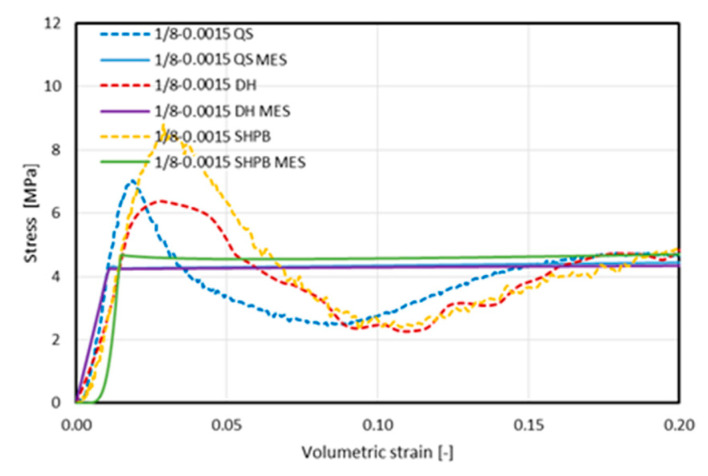
Summary of stress–volumetric strain characteristics obtained using the FEM–homogeneous model (FEM) and experimentally (exp) [21] at a quasi-static load (QS) strain rate of 8.3 × 10^−2^ 1/s, at a dynamic load (DH) strain rate of 3.8 × 10^2^ 1/s), and using a Split Hopkinson Pressure Bar (SHPB) at a strain rate of 3.3 × 10^3^ 1/s of the sample 1/8-5052-0.0015 for a strain limited to <0;0.2> scope.

**Table 1 materials-16-02211-t001:** Geometrical parameters of the studied honeycomb structures.

No.	Type	Declared * Cell Size *S* [mm]	Declared * Wall Thickness *t* [mm]	Height *T* [mm]
1	1/8-5052-0.0007	3.1750	0.01778	10
2	1/8-5052-0.001	3.1750	0.02540	10
3	1/8-5052-0.0015	3.1750	0.03810	10
4	3/16-5052-0.0007	4.7625	0.01778	10
5	3/16-5052-0.001	4.7625	0.02540	10
6	3/16-5052-0.0015	4.7625	0.03810	10
7	1/4-5052-0.001	6.3500	0.02540	10

* data provided by the producer’s catalogue [21].

**Table 2 materials-16-02211-t002:** List of characteristic dimensions obtained by measuring actual structures.

Measured Value	1/4-5052-0.001	3/16-5052-0.0007	3/16-5052-0.001	3/16-5052-0.0015	1/8-5052-0.0007	1/8-5052-0.001	1/8-5052-0.0015
R [mm]-bending radius of a single wall	0.27	0.22	0.28	0.39	0.19	0.26	0.37
d_theor_ [mm] theoretical wall width *	3.67	2.75	2.75	2.75	1.83	1.83	1.83
d_1_ [mm] single-wall width	3.57	3.02	3.06	3.14	2.19	2.11	2.14
d_2_ [mm] double-wall width	3.84	2.72	2.62	2.53	1.63	1.75	1.71

* provided in the manufacturer’s catalog card [21].

**Table 3 materials-16-02211-t003:** Comparison of the mechanical properties of aluminum Al 5052-H39, a foil composed of Al 5052-H39, and a fragment of a honeycomb core composed of that foil.

Parameter	Symbol	Unit	5052-H39 [23]	Foil 5052-H39 [24]	Core Fragment [24]
Density	ρ	kg/m^3^	2.7 × 10^3^	2.7 × 10^3^	2.7 × 10^3^
Elastic modulus	E	GPa	70.00	56.53	37.92
Yield stress	$R_{e}$	MPa	325.0	227.5	220.6
Tensile strength	$R_{m}$	MPa	330.0	248.2	234.4
Elongation at break	$\varepsilon_{u}$	%	4.0	1.6	4.7

**Table 4 materials-16-02211-t004:** Measurement results of the volume of air limited by walls in the real models of the samples.

	Graphical Representation	Wall Thickness *t* [mm]	Cell Size *S* [mm]	Closed Air Volume *V_a_* [mm^3^]
1/4-5052-0.001	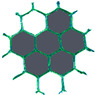	0.02540	6.3500	2264.5
3/16-5052-0.0007	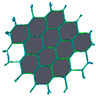	0.01778	4.7625	2556.8
3/16-5052-0.001	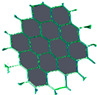	0.02540	4.7625	2702.0
1/8-5052-0.0015	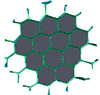	0.03810	4.7625	2744.9
1/8-5052-0.0007	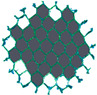	0.01778	3.1750	3426.4
1/8-5052-0.001	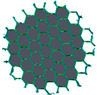	0.02540	3.1750	3548.8
1/8-5052-0.0015	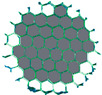	0.0381	3.175	3662.1

**Table 5 materials-16-02211-t005:** Parameters of the air enclosed inside the cores, according to the assumptions of the AIRBAG_SIMPLE_AIRBAG_MODEL [27].

Parameter	Symbol	Unit	Value
Density	ρ	kg/m^3^	1.2
Specific heat at constant volume	$c_{v}$	J/(kg·K)	713.00
Specific heat at constant pressure	$c_{p}$	J/(kg·K)	1000.00
Ambient pressure	$P_{e}$	MPa	0.1
Temperature	T	K	293
Area of holes	$A_{o}$	m^2^	0.004

**Table 6 materials-16-02211-t006:** List of the basic parameters of the examined structures.

Parameter	$\mathrm{Plateau}\;\mathrm{Stress}\;\sigma_{pl}$ [MPa]	$\mathrm{Elastic}\;\mathrm{Modulus}\;E_{u}$ [MPa]	$\mathrm{Critical}\;\mathrm{Stress}\;\sigma_{cr}$ [MPa]	$\mathrm{Relative}\;\mathrm{Volume}\;\mathrm{of}\;\mathrm{Total}\;\mathrm{Compaction}\;V_{F}$ [-]
1/4-5052-0.001	0.99 (−8.8%)	261.16 (−6.5%)	1.42 (9.6%)	0.28 (9.7%)
3/16-5052-0.0007	0.95 (3.1%)	168.22 (−6.7%)	1.60 (−6.0%)	0.28 (−7.7%)
3/16-5052-0.001	1.42 (3.4%)	247.73 (5.8%)	2.22 (−1.4%)	0.29 (−7.4%)
1/8-5052-0.0015	2.28 (2.6%)	397.34 (4.7%)	3.54 (10.6%)	0.21 (27.6%)
1/8-5052-0.0007	1.49 (2.6%)	251.34 (3.8%)	1.72 (9.5%)	0.27 (−8.0%)
1/8-5052-0.001	2.70 (−2.3%)	474.06 (8.8%)	4.11 (3.1%)	0.24 (7.7%)
1/8-5052-0.0015	4.41 (1.3%)	591.11 (−1.9%)	6.82 (3.0%)	0.21 (30.0%)

## Data Availability

The data presented in this study are available on request from the corresponding author.

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
