# Peer review of "A Study of Aluminum Honeycomb Structures under Dynamic Loading, with Consideration Given to the Effects of Air Leakage"

_materials, 2023, doi:10.3390/ma16062211_

Round 1

Reviewer 1 Report

The article is based on the latest findings in this area of research. It is very valuable from the point of view of deformation modelling for different types of structures. 

In the articles I miss the reference to various applications focused in the automotive industry and their subsequent ev. use which are mentioned e.g. in the references On Aluminum Honeycomb Impact Attenuator Designs for Formula Student Competitions by Quoc, P.M. et al . So please review the literature and put other relevant citations with a focus on impact attenuator applications.

Author Response

Dear Reviewer,

Thank You very much for Your valuable comments and for the ability to improve or work.

Please find our answers below.

We added the info on applications of honeycombs in automotive and military industry (marked in yellow in the text).

Reviewer 2 Report

Language/Formatting

1)     The whole manuscript must be improved to a good standard of English.

2)     The whole manuscript needs to be reorganized as current version is difficult for the first-time reader to keep up with the content of the work. Substantial improvement is deemed required

3)     All Figures should be properly introduced in the paragraph rather than mentioning it in bracket.

4)     Include preamble for section 4 to introduce its subsections

5)     Subheading titles for 4.1 – 4.4 need to be written in a consistent way/style.

6)     Reference 19 cited in the Table 3 got error.

Content

7)     Abstract needs to be rewritten. It should include the statement of the problem or central issue that to be addressed, research/analytical method, qualitative/quantitative results that reflect the major finding of the study and significance of the study.

8)     An introduction section needs to be further improved. The general and specific background of the works must be further elaborated in order to clearly show the importance of the study.

9)     Specific objectives of the study need to be clearly mentioned in the introduction section.

10)  Kindly include more recent literature that relates to the work to justify the relevance of the proposed work.

11)  Include references for the statement line 77 – 88

12)  The MAT_HONEYCOMB and MAT_MODIFIED_HONEY_COMB should be first introduced as a one of the material models to represent the metallic honeycomb in LS-DYNA. A literature review needs to be added to assess the relevancy/effectiveness of using these models.

13)  Avoid any equations in introduction. Equations should be described in the methodology section if it is employed in the study

14)  Figure 2 process flow is unclear, kindly use a proper block of flowchart process to better represent the research methodology scheme.

15)  The methodology should be properly described in the manuscript

a.      Include information on the mesh convergence analysis

b.  All the FE model No of elements, element types, properties, boundary condition and loading should be described in this section instead

c.      Dynamic loading model/response should be clearly described.

16)  Result and Discussion section should only focus on the results and the explanation of the results. All the methods and analysis setup should be explained in the methodology section instead.

17)  What is the considered approach to initially validate the FE model so that it is good enough for any subsequent analysis.

18)  How can the FE model is updated if there is any discrepancy against its experimental counterpart.

19)  Conclusion should reflect the specific objectives of the work and also the title of the manuscript. Any further discussion/finding should be avoided in the conclusion section. Discussion of the findings should be discussed in the discussion section instead.

Author Response

Dear Reviewer,

Thank You very much for Your valuable comments and for the ability to improve or work.

Please find our answers below (the changes made in the text in accordance to Your comments were marked in blue)

1. The whole manuscript must be improved to a good standard of English.

The paper is now checking in MDPI language service.

2. The whole manuscript needs to be reorganized as current version is difficult for the first-time reader to keep up with the content of the work. Substantial improvement is deemed required

Thank You for this comment. We reorganized the manuscript. First we added Section 3 in which we describe the models preparation and validation (sections 4.1 and 4.3 from the first version of the paper). In Section 4 we put the results of FE analyses of both types of models and we also added the discussion previously shown in Conclusions.

3. All Figures should be properly introduced in the paragraph rather than mentioning it in bracket.

The paper was improved in accordance to this comment.

4. Include preamble for section 4 to introduce its subsections

We added short preambles to Section 3 and 4 – see also answer to comment 2.

5. Subheading titles for 4.1 – 4.4 need to be written in a consistent way/style.

Thanks, the titles were prepared on the base of MDPI template In a correct way.

6. Reference 19 cited in the Table 3 got error.

Thanks, we checked and corrected it.

7. Abstract needs to be rewritten. It should include the statement of the problem or central issue that to be addressed, research/analytical method, qualitative/quantitative results that reflect the major finding of the study and significance of the study.

The abstract  has been rewritten. The main problem is pointed out now. Some sentences regarding the research process and obtained results are added.

8. An introduction section needs to be further improved. The general and specific background of the works must be further elaborated in order to clearly show the importance of the study.

The introduction has been improved.

9. Specific objectives of the study need to be clearly mentioned in the introduction section.

The main goal was set in the text (obtaining valuable results in wide strain rate range with no need to conduct an experimental investigation).

10. Kindly include more recent literature that relates to the work to justify the relevance of the proposed work.

Articles 5-8 has been added to the refferences

11. Include references for the statement line 77 – 88

As we understand the comment is similar to comment 11 – please see the answer there.

12. The MAT_HONEYCOMB and MAT_MODIFIED_HONEY_COMB should be first introduced as a one of the material models to represent the metallic honeycomb in LS-DYNA. A literature review needs to be added to assess the relevancy/effectiveness of using these models.

Some references has been added.

13. Avoid any equations in introduction. Equations should be described in the methodology section if it is employed in the study

The equations were described in the section describing the homogenous model. See also answer to comment 2.

14. Figure 2 process flow is unclear, kindly use a proper block of flowchart process to better represent the research methodology scheme.

Figure 2 was improved.

15. The methodology should be properly described in the manuscript

a)Include information on the mesh convergence analysis

The information has been added

b) All the FE model No of elements, element types, properties, boundary condition and loading should be described in this section instead

The information has been added.

c) Dynamic loading model/response should be clearly described.

Boundary conditions, loadings has been described on fig. 4 and fig. 14

16. Result and Discussion section should only focus on the results and the explanation of the results. All the methods and analysis setup should be explained in the methodology section instead.

We corrected the paper due to this comment, please see answer to comment 2.

17. What is the considered approach to initially validate the FE model so that it is good enough for any subsequent analysis.

Initial validation was carried out by comparing stress and volumetric strain characteristics. Further analysis will concern the comparison of internal energy and work (FEM) of external forces (experiment)

18. How can the FE model is updated if there is any discrepancy against its experimental counterpart.

No significant differences were noted when comparing static test results. In the case of differences in dynamic results, an appropriate leakage coefficient that could be used for the entire strain rate range was sought

19. Conclusion should reflect the specific objectives of the work and also the title of the manuscript. Any further discussion/finding should be avoided in the conclusion section. Discussion of the findings should be discussed in the discussion section instead.

We moved the discussion to the Results & discussion Section.

Reviewer 3 Report

In the presented study the authors proposed and verified the method of numerical modelling of honeycomb materials using simplified Y-element for material properties description, which are further use in homogenized model. In addition the influence of pressure and leakage of air entrapped in the honeycomb cells was considered. This is beneficial to increase the understand of honeycomb structures. However, there are still some problems in this paper. The following suggestions are for the author's reference:

1)      As can be seen in Figures 16 and 17, the homogeneous model does not have the first peak, what is the reason for this? How can the accuracy of the homogeneous model be verified?

2)      How is the air content in a honeycomb structure determined? Do different air contents affect the results?

3)      What is the purpose of testing different scale results?

Author Response

Dear Reviewer,

Thank You very much for Your valuable comments and for the ability to improve or work.

Please find our answers below

1. As can be seen in Figures 16 and 17, the homogeneous model does not have the first peak, what is the reason for this? How can the accuracy of the homogeneous model be verified?

When defining the stress-strain characteristics for the honeycomb mat model, the first peak should be omitted. Otherwise, individual layers of elements will behave as separate honeycombs as below (attached file).

Initial validation was carried out by comparing stress and volumetric strain characteristics. Further analysis will concern the comparison of internal energy and work (FEM) of external forces (experiment)

2. How is the air content in a honeycomb structure determined? Do different air contents affect the results?

The air inside the core has been defined with the airbag formulation. The basic assumption was to include air as simply as possible. Changes in the leakage rate caused noticeable differences in the results obtained

3. What is the purpose of testing different scale results?

The tests of a single element Y were carried out to check the usefulness of the test results of a simple model for use in a homogeneous model. Tests of the homogeneous model at different strain rates were carried out to evaluate the possibility of using it to model large structures in the interests of low numerical cost.

Round 2

Reviewer 2 Report

Thank you very much on the revised version of the manuscript and the responses made to the comments. The manuscript has been greatly improved compared to its first version.

1.      Please take note that there is still an error in referencing in Table 3 that shows as following.

Foil 5052-H39 [24Error! Refer-ence source not found.]

Kindly correct it accordingly.

2.      Figure 15 and 16, it is suggested to properly label the caption with the title for each of the subfigures rather than mentioning them in the main caption.

Author Response

Thank You once again for Your comments. 

We upgraded the paper as follows:

  1. We corrected the reference in table 3 - we hope it is correct. It was not visible for us in Word - it is probably because of the transfer from Word to PDF and the automatic formatting of literature numbers.
  2. We change figures 15 and 16 in accordance to Your comment.

The changes are marked in purple.

Thanks again.